# Bacterial clustering amplifies the reshaping of eutrophic plumes around marine particles: A hybrid data-driven model

**George E. Kapellos**[1,2]*, **Hermann J. Eberl**[3], **Nicolas Kalogerakis**[4], **Patrick S. Doyle**[1], **Christakis A. Paraskeva**[2]

**1** Department of Chemical Engineering, Massachusetts Institute of Technology, Cambridge, Massachusetts, United States of America, **2** Department of Chemical Engineering, University of Patras, Rion Achaia, Greece, **3** Department of Mathematics and Statistics, University of Guelph, Ontario, Canada, **4** School of Chemical and Environmental Engineering, Technical University of Crete, Chania, Greece

* kapellos@mit.edu

**Data Availability Statement:** All data needed to evaluate the conclusions in the paper are present in the manuscript, cited sources, and Supporting information files.

## Abstract

Multifaceted interactions between marine bacteria and particulate matter exert a major control over the biogeochemical cycles in the oceans. At the microbial scale, free-living bacteria benefit from encountering and harnessing the plumes around nutrient-releasing particles, like phyto-plankton and organic aggregates. However, our understanding of the bacterial potential to reshape these eutrophic microhabitats remains poor, in part because of the traditional focus on fast-moving particles that generate ephemeral plumes with lifetime shorter than the uptake timescale. Here we develop a novel hybrid model to assess the impacts of nutrient uptake by clustered free-living bacteria on the nutrient field around slow-moving particles. We integrate a physics-based nutrient transport model with data-derived bacterial distributions at the single-particle level. We inferred the functional form of the bacterial distribution and extracted parameters from published datasets of *in vitro* and *in silico* microscale experiments. Based on available data, we find that exponential radial distribution functions properly represent bacterial microzones, but also capture the trend and variation for the exposure of bacteria to nutrients around sinking particles. Our computational analysis provides fundamental insight into the conditions under which free-living bacteria may significantly reshape plumes around marine aggregates in terms of the particle size and sinking velocity, the nutrient diffusivity, and the bacterial trophic lifestyle (oligotrophs < mesotrophs < copiotrophs). A high potential is predicted for chemotactic copiotrophs like Vibrio sp. that achieve fast uptake and strong clustering. This microscale phenomenon can be critical for the microbiome and nutrient cycling in marine ecosystems, especially during particulate blooms.

## Author summary

Recent lines of evidence highlight the pronounced impact of slow-moving particles on the oceanic carbon cycle and associated ecosystem functions (e.g., $CO_2$ removal, oxygenation,

**Funding:** This work received funding from the European Union's Horizon 2020 research and innovation programme under a Marie Skłodowska-Curie grant agreement (No. 741799, "OILY MICROCOSM" to GEK). The funders had no role in study design, data collection and analysis, decision to publish, or preparation of the manuscript.

**Competing interests:** The authors have declared that no competing interests exist.

acidification). In contrast to fast ones, slow-moving particles generate large and persistent eutrophic plumes of dissolved organic matter (DOM) and host intense interactions between surface-attached and free-living bacteria. However, significant aspects of the multifaceted biochemical coupling in these eutrophic microhabitats remain largely unexplored. Here we elucidate the potential of free-living bacteria to reshape microscale eutrophic plumes with a particle-level model that combines a physics-based description of the chemical field with a data-based description of bacterial clusters. This hybrid framework captures salient features and impacts of bacterial clustering in a simple and efficient manner, while bypassing inherent uncertainties of more sophisticated bacterial transport models. Our computational analysis delineates the conditions and types of particles, bacteria, and DOM for which plume reshaping is expected to be important.

## Introduction

In vast oligotrophic oceans, organic particles offer oases full of resources to marine bacteria. Multifaceted physical and biochemical interactions between the bacteria and particulate organic matter (POM) underpin the health and essential functions of oceanic ecosystems by modulating the cycles of carbon and inorganic elements (N, P, Fe, S), the marine primary production and food webs, the removal of atmospheric $CO_2$, the levels of seawater oxygenation and acidification, and the efficiency of the biological carbon pump (i.e., vertical transport and storage of organic carbon into the deep ocean) [1–3]. Advanced mechanistic understanding of microscale oceanic processes between microorganisms and POM is a major enabler towards a sustainable development in marine environments.

Marine particles generate microscale eutrophic plumes of dissolved organic matter (DOM) with products from the enzymatic hydrolysis of particulate ingredients and the metabolic activities of microbial particle-dwellers [4–7]. The volume of a plume may be 10–100 times the particle volume [8,9], with nutrient concentrations from one to three orders of magnitude higher than ambient levels [10–12], thereby offering a unique nutritional opportunity to planktonic microbes. Chemotactic bacteria, in particular, may actively track plumes by detecting fluctuations in the concentration of the emitted chemical cues. However, plume tracking can be successful only if the chemotaxis timescale is shorter than the plume lifetime. The chemotaxis timescale is determined by the bacterial systems of chemosensing (i.e., palette and thresholds of detected solutes) and seascape navigation (i.e., tuning of swimming speed and direction) [13,14]. The plume lifetime may range from several seconds to tens of minutes depending on the size and velocity of the particle, the flow regime, the mechanisms of nutrient release, the spreading due to advection and diffusion, and the consumption by planktonic bacteria [9,15,16].

Microscale experiments have demonstrated the capacity of chemotactic marine bacteria to tune their navigation mode and develop high swimming speeds (10–1000μm/s) so as to track and rapidly colonize both ephemeral and persistent plumes [17–23]. For instance, the prototypical plume trackers *Shewanella putrefaciens* and *Pseudoalteromonas haloplanktis* have been found to successfully pursue motile algae [18], to form microzones around nutrient-releasing beads [19], and to massively accumulate within plumes of algal exudates [20]. Similar observations have been made with computer simulations [24–29]. Efficient plume trackers benefit from harnessing eutrophic plumes and secure high growth rates in the presence of organic particles, plumes and associated nutrient gradients. Plume tracking and feeding play a critical role in the response of local microbiomes to sporadic or seasonal releases of POM, like after

phytoplankton blooms [30] and oil spills [31]. For example, the Deepwater Horizon oil spill caused a subsea hydrocarbon plume that stimulated the growth of bacteria in the Oceanospirillales order with genes for chemotaxis and alkane degradation [32], thus indicating that these bacteria are capable of tracking and exploiting oil droplets.

Particle-based models provide a link between microbial-scale processes and ocean-level dynamics of carbon cycling and microbial growth. Significant modeling work has been done on the potential of particle-associated bacteria to subsidize eutrophic DOM plumes through the enzymatic hydrolysis of POM [33–36], and the concomitant chemotactic response of free-living bacteria to track the plumes [20,37]. However, the potential of free-living bacteria to reshape the plumes has been largely overlooked because of the traditional focus on fast-moving particles that create short-lived plumes and, also, due to a lack of simple mathematical descriptions of bacterial clustering [16]. Nonetheless, recent lines of evidence leverage the significance of suspended and slow-moving particles that generate large persistent plumes [38–41], amenable to transformation by planktonic bacteria when the uptake timescale is shorter than the plume lifetime. In that vein, we recently suggested that even uniformly distributed bacteria may substantially reshape nutrient plumes around individual phytoplankton and marine aggregates, when the particles sink slowly at less than 40 m/d [9]. Here, we develop a hybrid microscale model to quantitatively assess the impacts of bacterial clustering on the reshaping of eutrophic plumes around marine particles. The model formulation combines a physics-based description for the nutrient field with a data-based description for the bacterial distribution at the single-particle level.

## Results and discussion

### Bacterial microzones around marine particles

We inferred the functional form and parameters for the spatial distribution of marine bacteria around nutrient-releasing particles by mining information hidden in published data of *in vitro* and *in silico* microscale experiments [19,21,25,29]. By analyzing four independent datasets, we found that the concentration of free-living bacteria around the particles, $B$, is well-described by an exponential radial distribution function (RDF; Fig 1), expressed in dimensionless generic form as:

$$B(\mathbf{x}) = \beta_0 + \beta_m \mathbf{exp}\left[-(r-1)^n/d_s^n\right] \qquad (1)$$

Here, $\mathbf{x}$ is the position vector with reference to the particle center, $r = |\mathbf{x}|$ is the radial distance, and $\{\beta_0, \beta_m, d_s, n\}$ are the RDF parameters. For the dimensional analysis, the reference length is the particle radius, $\tilde{R}_P$, and the reference bacterial concentration is the ambient average value, $\tilde{B}_v^\infty$, that is measured over a volume of water much larger than the particle volume. The tilde (~) over a symbol denotes a dimensional quantity, whereas its absence denotes a dimensionless one.

The RDF exponent $n$ controls the shape of the distribution curve, with $n = 1$ for single exponential and $n = 2$ for Gaussian-like attenuation. The parameter $\beta_m$ is the peak concentration and denotes the maximum of the distribution at the particle surface, $d_s$ is the characteristic accumulation length and denotes the distance from the particle surface at which the excess bacterial concentration drops by 63%, that is $B(1 + d_s) - \beta_0 = 0.37\beta_m$, and $\beta_0$ is the renormalized ambient concentration at the particle scale. Using constrained nonlinear regression analysis, we extracted the RDF parameters given in Table 1 from data of microfluidic and computational experiments (Fig 1B; S1 Appendix). A metric for the degree of bacterial clustering around the particle is the hotspot index, $h_e$, which is defined as the ratio of the average

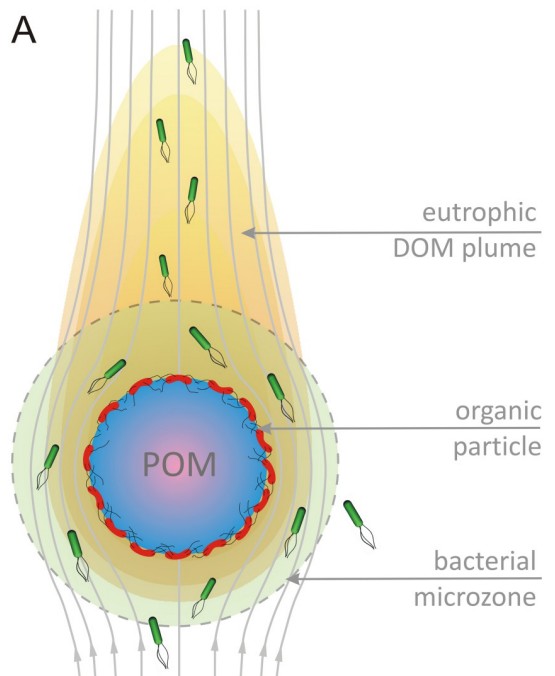
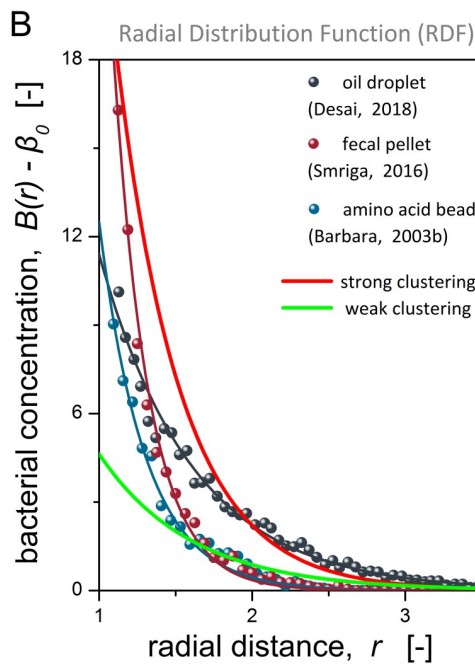

**Fig 1. Bacterial microzones. (A)** Conceptual illustration of the microzone model. The moving particle (POM) creates a comet-shaped eutrophic plume of dissolved organic matter (DOM) with particulate lysate and metabolites of surface-attached bacteria (red). The DOM plume attracts chemotactic bacteria (green), which act as point sinks and reshape the nutrient field. **(B)** Normalized radial distribution functions (RDF) for the excess bacterial concentration around nutrient-releasing particles, obtained by mining information from published data of *in vitro* and *in silico* microscale experiments (RDF parameters in Table 1).

bacterial concentration in a microzone around the particle over the ambient value. The hotspot index is in the range of $h_e \sim 2 - 4$ for sinking particles [20,26].

The observed bacterial accumulation in proximity to the particle (Fig 1) is attributed to biochemical and rheological mechanisms. High concentrations and sharp gradients of nutrients and infochemicals develop around the particle and attract chemotactic bacteria [14]. Furthermore, bacteria are entrained into the Darwin drift volume around the particle and get towed along the particle's path due to hydrodynamic interactions with the particle surface [37,42].

**Table 1. RDF parameters.** Datasets (1)-(4) are extracted with non-linear regression analysis of published data from microscale and *in silico* experiments (S1 Appendix), and datasets (5)-(6) are used for hypothesis testing in this work. All datasets are used with the RDF: $B(r) = \beta_0 + \beta_m exp\left[-(r-1)^n/d_s^n\right]$, and each dataset is normalized by a different average bacterial concentration ($\tilde{B}_v^\infty$).

| No. | Source | $\tilde{R}_P$ [μm] | $\tilde{B}_v^\infty$ [cells/mL] | $n$ | $\beta_0$ | $\beta_m$ | $d_s$ | $R_M$ | $h_e$ |
|---|---|---|---|---|---|---|---|---|---|
| 1 | oil droplet, computational experiment (Desai, 2018) | 20 | $3.1 \times 10^9$ | 1 | 0.06 | 11.4 | 0.61 | 3.9 | 1.1 |
| 2 | amino acid bead, microscale experiment (Barbara, 2003b) | 32 | $0.7 \times 10^9$ | 1 | 0.07 | 12.5 | 0.30 | 2.5 | 1.5 |
| 3 | fecal pellet, microscale experiment (Smriga, 2016) | 80 | $3.5 \times 10^6$ | 1 | 0.31 | 27.4 | 0.23 | 2.2 | 3.0 |
| 4 | algal cell, computational experiment (Bowen, 1993) | 20 | $1.0 \times 10^6$ | 1 | 0.95 | 68.0 | 1.15 | 8.5 | 3.2 |
| 5 | strong clustering, hypothetical scenario (this work) | – | $10^6$–$10^7$ | 1 | 1.00 | 23.9 | 0.42 | 3.3 | 2.8 |
| 6 | weak clustering, hypothetical scenario (this work) | – | $10^6$–$10^7$ | 1 | 1.00 | 4.63 | 0.60 | 3.3 | 1.6 |

The microzone radius is estimated as $R_M = 1 + d_s \left[\ln(10\beta_m)\right]^{1/n}$ and corresponds to an excess bacterial concentration 10% above the baseline ($B - \beta_0 = 0.1$). The hotspot index, $h_e$, is calculated with Eq (6). The microzone radius and the hotspot index are metadata, which are calculated once the RDF parameters have been determined by nonlinear regression. The low values of $\beta_0$ are attributed to the small observation windows used in the sources of those data (see details in S1 Appendix).

Bacterial microzones are sheltered from oceanic turbulent interference, when the particle is smaller than the Kolmogorov microscale that is about 1-6mm in rough seas with high energy dissipation rate ($\sim 10^{-2}$ cm$^2$/s$^3$) [43], and hence promote rich chemical exchanges between the diverse populations of particle-associated and free-living bacteria [4]. The ecological significance of the microzone concept is well-established for algal cells ("phycosphere") [22] and marine snow [44], and recently extended to microplastics ("plastisphere") [45] and eukaryotic protists ("mucosphere") [46]. Nonetheless, the availability of data for the spatial organization of bacterial microzones is still limited by technical challenges in the quantitative characterization of such microscopic volumes around moving particles. Here, we took a first step towards unifying observations from different settings under a simple, but fundamental to natural phenomena, functional family.

## Chemotaxis and RDF parameters

Bacterial chemotaxis is included in Eq (1) through the values of the RDF parameters. Although rather limited and from disparate sources/systems, the available datasets support the following observations. First, the good fit to the data by simple exponential functions with maximum at the particle surface suggests that there is no inhibition by the chemoattractant or any competing gradients from other chemoattractants or repellants that could result in more complex patterns, such as band formation [19].

Furthermore, the RDF parameters from the first three datasets in Table 1, show that an increase of the normalized peak concentration ($\beta_m$) is accompanied by a decrease of the accumulation length ($d_s$). This trend may be attributed to two different chemotactic mechanisms. For chemotactic bacteria with run-and-tumble motility, lower swimming speeds result in elevated peak concentrations and shorter accumulation lengths, i.e., thinner and denser microzones (Fig 2). In this mode of motility, any gain in the run of long distances by swimming fast, comes at the cost of reduced ability to maneuver and focus close to confined nutrient sources [14]. By contrast, for chemokinetic bacteria with run-reverse motility and modulation of their swimming speed, the opposite trend has been reported [47]. That is, the microzone becomes thinner and denser as the average swimming speed increases because chemokinetic bacteria fine tune their speed and, thus, increase their ability to maneuver as they approach a chemoattractant source.

Although the above trend for $\{\beta_m, d_s\}$ is confirmed by simulations of *Bowen et al.* [25] (see Fig 2), their parameter values stand out due to differences in the underlying mechanisms and experimental setups (Table 1 and S1 Appendix). In particular, the high values of normalized peak and ambient concentrations ($\beta_m$ and $\beta_0$) are partly attributed to the large observation window used by *Bowen et al.* [25]. Moreover, the striking difference in the accumulation length ($d_s$), which is 2–5 times larger for the algal cell than other datasets, is rooted in chemotactic features (swimming speed, average run time, chemoreceptor saturation) that favor the formation of wider microzones by bacteria with run-and-tumble motility. For instance, in the individual-based simulations of *Bowen et al.* [25], bacteria run about 40μm between tumbles (i.e., one particle diameter). The respective average run length is only 6μm in the simulations of *Desai et al.* [29].

The role of the RDF exponent ($n$) is somewhat more intricate. For $n = 1$, the derivative of the exponential RDF at the particle surface is $B'(1) = \beta_m / d_s$ and, as discussed above, sharp RDFs (high $\beta_m$, low $d_s$) correspond to thin and dense microzones. For $n > 1$, the derivative becomes $B'(1) = 0$ and implies relation to wide microzones. For $n < 1$, the derivative becomes $B'(1) \rightarrow \infty$ and suggests association with very confined microzones. In the literature, the value of $n = 1$ is established [47], as it appears in the steady state solution of the Keller-Segel model

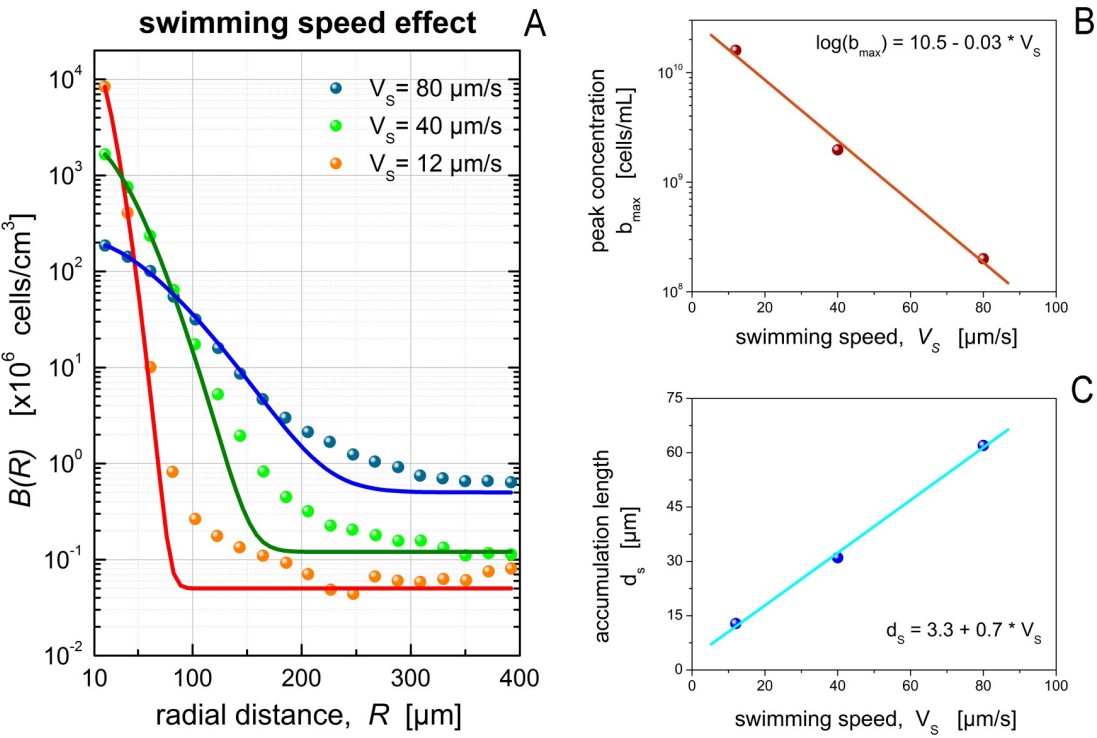

**Fig 2. Correlation of bacterial swimming to RDFs.** Impact of the bacterial swimming speed on the accumulation of marine bacteria with run-and-tumble motility around nutrient-exuding algae. (A) The data points originate from individual-based simulations by *Bowen et al.* (Fig 2A in [25]) and the continuous lines represent optimal fit of our exponential RDF (see section S1.5 in S1 Appendix). The bacterial peak concentration (B) and the chemotactic accumulation length (C) are negatively and positively correlated to the bacterial swimming speed, respectively, for this mode of motility.

of chemotaxis. In this work, we examined Gaussian and fractional RDF exponents and found that fractional exponents provide best-fit in certain cases (S1 Appendix). For example, the optimal exponent is $n = 4/5$ for a synthetic dataset and the oil droplet, and $n = 3/5$ for the fecal pellet. However, for consistency, we used the near-optimal integer exponent $n = 1$ throughout the subsequent analysis of plume quenching.

Finally, our analysis of bacterial RDFs from computer simulations by *Bowen et al.* [25], suggests that the RDF exponent ($n$) depends strongly on the interplay between chemoattractant exudation and bacterial chemotaxis (Fig U in S1 Appendix). The chemoattractant exudation rate affects the thickness of the concentration boundary layer around the particle, while the chemotactic response weighs in the ability of the bacteria to maneuver and focus within the nutrient-rich layer. Thus, under a given flow regime, thicker boundary layers may support thicker and denser microzones (high $\beta_m$, high $d_s$). Moving forward, a systematic correlation between RDF parameters and underlying mechanisms presents an exciting avenue for future research, as complex cell-scale processes are parameterized in simple models on the particle scale.

## A hybrid model of microscale plume (re)shaping

The chemical field around marine organic particles is shaped by the interplay between advection, diffusion and microbial transformation. The relative importance of these processes is quantified by the dimensionless Péclet and Damköhler numbers, which can be expressed in terms of

fundamental timescales as $Pe = \tilde{\tau}_D/\tilde{\tau}_A$ and $Da = \tilde{\tau}_D/\tilde{\tau}_U$, respectively. Here, $\tilde{\tau}_D = \tilde{R}_P^2/\tilde{D}_{Av}$ is the diffusion timescale, $\tilde{\tau}_A = \tilde{R}_P/\tilde{v}_\infty$ is the advection timescale, $\tilde{\tau}_U$ is the uptake timescale, $\tilde{v}_\infty$ is the ambient water velocity, and $\tilde{D}_{Av}$ is the nutrient diffusivity. In the particle frame of reference, the advection-diffusion-bioreaction equation that describes the formation of quasi-steady DOM plumes around the particle can be expressed in dimensionless generic form as follows:

$$Pe\mathbf{v} \cdot \nabla C = \nabla^2 C - DaB(\mathbf{x})a(\mathbf{x})C \qquad (2)$$

Here, $C$ is the nutrient concentration, and $\mathbf{v}$ is the fluid velocity. For the dimensional analysis, the concentration $\tilde{C}_{ref}$ at the particle surface (e.g., solubility) is the reference concentration, and the ambient water velocity $\tilde{v}_\infty$ is the reference velocity.

The product $a(\mathbf{x})C(\mathbf{x})$ is the nutrient uptake rate per single cell and the affinity factor, $a(\mathbf{x})$, accounts for nonlinear effects of physical and biochemical stressors, such as saturation, inhibition and multi-substrate limitation. For instance, a = 1 for unsaturable uptake and $a = K_S/(K_S + C)$ for Michaelis-Menten kinetics [48], where $K_S$ is the dimensionless half-saturation constant. Furthermore, the product $B(\mathbf{x})C(\mathbf{x})$ represents the encounter rate between bacteria and DOM and, accordingly, the average nutrient exposure of a bacterial population in a specified volume of seawater (e.g., the microzone volume, $V_M$) is calculated as:

$$c^* = \int_{V_M} B(\mathbf{x})C(\mathbf{x})dV \Big/ \int_{V_M} B(\mathbf{x})dV \qquad (3)$$

As shown in Fig 3, the amplification in nutrient exposure due to bacterial clustering around slow-moving particles is particularly pronounced. By contrast, fast-sinking particles create

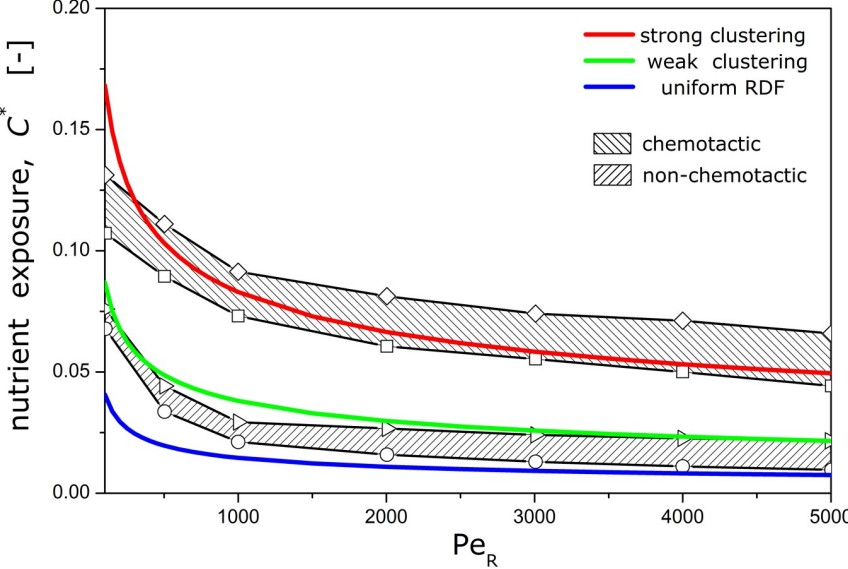

**Fig 3. Nutrient exposure.** The bacterial microzone model captures *in silico* observations for the nutrient exposure of free-living bacteria in the undisturbed nutrient field (Da = 0) around a sinking particle. The solid lines correspond to bacterial distributions described by the exponential RDFs used in this work with red color for strong clustering, green color for weak clustering, and blue for a uniform distribution. The shaded areas correspond to results from computer simulations by *Desai et al.* [37] for chemotactic bacteria with (◇) or without (□) hydrodynamic interactions, and non-chemotactic bacteria with (△) or without (○) hydrodynamic interactions.

slender eutrophic plumes, characterized by high Péclet numbers, and offer reduced nutrient exposure to free-living bacteria, albeit substantially higher than ambient levels ($c^* > 0.001$).

In terms of fundamental timescales, plume colonization is plausible if the bacterial chemotaxis is faster than plume dissipation due to advection and diffusion, $\tilde{\tau}_C < \tilde{\tau}_{PLM}$, and plume depletion due to uptake, $\tilde{\tau}_C < \tilde{\tau}_U$. The undisturbed plume lifetime, $\tilde{\tau}_{PLM} = min\{\tilde{\tau}_A, \tilde{\tau}_D\}$, may range from several seconds to tens of minutes [15,16], while the chemotaxis timescale, $\tilde{\tau}_C$, is on the order of a few seconds [13,14]. Consequently, chemotactic marine bacteria may always achieve a degree of clustering in the presence of particles, plumes and associated chemical gradients. One step further, plume reshaping is expected to be significant when $\tilde{\tau}_C < \tilde{\tau}_U < \tilde{\tau}_{PLM}$ and, in accordance with dimensional analysis, the degree of reshaping depends on the Péclet and Damköhler numbers. In the special case of $\tilde{\tau}_C \sim \tilde{\tau}_U$, the steady-state solution of Eq (2) is null and the transient fully-coupled analysis of nutrient and bacteria transport is required [49,50]. The uptake timescale depends on the bacterial abundance and the nutrient affinity, $\tilde{\tau}_U = (\tilde{B}_v^\infty \tilde{\alpha}_S)^{-1}$. The average bacterial abundance, $\tilde{B}_v^\infty$, ranges from $10^4$ *cells/mL* in the deep ocean to $10^7$ *cells/mL* in coastal waters [51], and the bacterial affinity for organic and inorganic nutrients, $\tilde{\alpha}_S$, ranges from tens of femtoliters up to a few picoliters per second per cell (Table 2 in [9]). In the presence of organic particles, the bacterial abundance and nutrient affinity are expected on the high end of their ranges, that is $\tilde{B}_v^\infty = 10^6 - 10^7$ *cells/mL* and $\tilde{a}_S = 1 - 10 \, pL/(cell \cdot s)$. Hence the uptake timescale may range from tens to hundreds of seconds [16,28]. For uniformly distributed bacteria, we have recently shown that the timescale condition $\tilde{\tau}_U < \tilde{\tau}_{PLM}$ is satisfied if Pe/Da<100 and Da $> 10^{-4}$ [9]. Given the rapid chemotactic response of marine bacteria ($\tilde{\tau}_C < \tilde{\tau}_U$) [14,28], here we examine the effects of bacterial clustering and uptake strength on the pattern and characteristic metrics of the nutrient field, under realistic conditions for marine aggregates and phytoplankton.

## Predicted impact of bacterial clustering on plume reshaping

We considered two levels of clustering, strong and weak, described with the exponential RDF of Eq (1) and the parameters listed in Table 1. Both distributions correspond to the same microzone radius ($R_M = 3.3$), but different hotspot index ($h_e = 2.8$ for strong clustering and $h_e = 1.6$ for weak clustering). The selection of the RDF parameters was based on two criteria. First, to capture the range of values extracted from published data (Fig 1) for the bacterial peak concentration ($\beta_m$) and the chemotactic precision length ($d_s$). Second, to match the trend and spectrum of the simulation data provided by *Desai et al.* [37] for the exposure of bacteria to the undisturbed nutrient field (Da = 0) around a sinking particle (Fig 3). These two clustering models drill through the parameter space of $\{\beta_m, d_s\}$ and are indicative, rather than fully representative.

Furthermore, we considered two levels of uptake: normal with $\tilde{\tau}_U = 1000s$, and fast with $\tilde{\tau}_U = 100s$ (upregulated). Normal uptake represents typical conditions of a bacterial population with background concentration $\tilde{B}_v^\infty = 10^6$ *cells/mL* and nutrient affinity $\tilde{\alpha}_S = 1 \, pL/(cell \cdot s)$. Fast uptake may well be achieved by bacterial populations of higher abundance (e.g., during particulate blooms [30]) and/or higher uptake affinity (e.g., associated with acclimated copiotrophic bacteria uptaking small nutrient molecules [52]). For reference, *Jackson* [16] considered an uptake timescale of 637s, while *Taylor & Stocker* [28] considered $\tilde{\tau}_U = 200s$.

Here, we examine the effects of the bacterial distribution and activity on plume reshaping for a wide range of particle size (ESD) and velocity (SV) of sinking marine aggregates. Based on the available experimental datasets, we distinguish three characteristic ranges of particle sizes (Fig 4): small (ESD<0.3mm), medium (0.3mm<ESD<0.8mm), and large (ESD>0.8mm); and three ranges of sinking velocities: low (SV<20m/d), moderate (20m/d<SV<100m/d), and

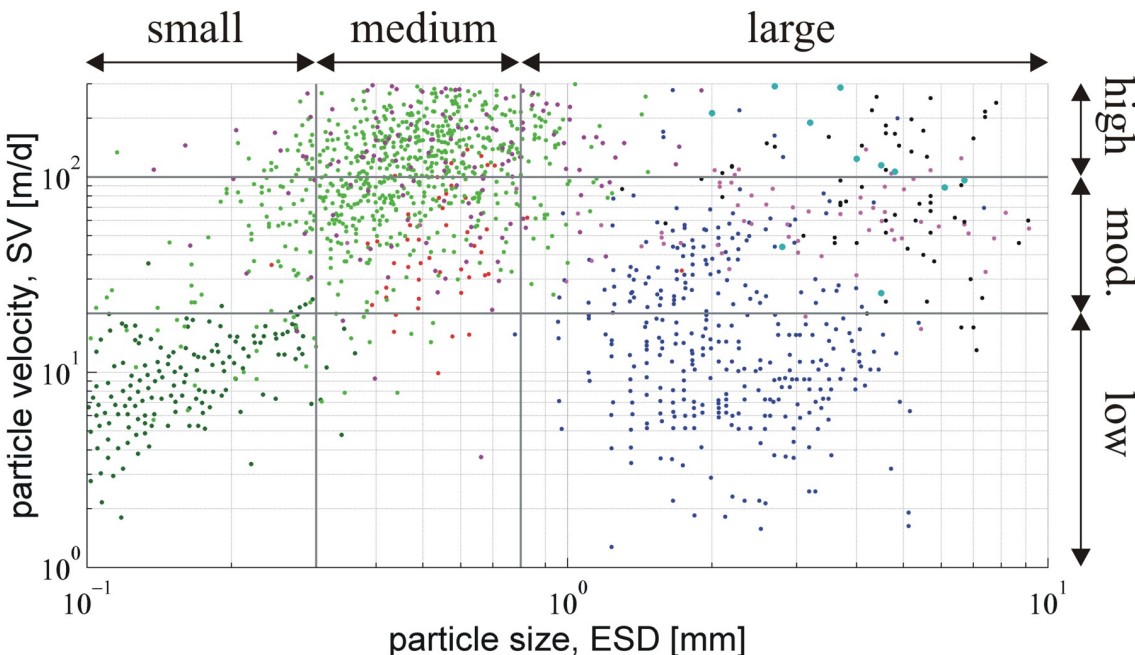

**Fig 4. Classification of sinking marine particles.** The points represent experimental data for the sinking velocity (SV) and the equivalent sphere diameter (ESD) of individual aggregates. The vertical and horizontal straight lines delimit the boundaries between particle classes. We distinguish three classes with respect to particle size: small (ESD<0.3mm), medium (0.3mm<ESD<0.8mm), and large (ESD>0.8mm); and three classes with respect to sinking velocity: low (SV<20m/d), moderate (20m/d<SV<100m/d), and high (100m/d<SV<300m/d). Source of experimental data: [53] dark green, [54] green, [55] purple, [56] red, [57] blue, [58] magenta, [59] dark cyan, [60] black.

high (100m/d<SV<300m/d). To facilitate the analysis, we identify three representative particle classes for which plume reshaping is meaningful (>10% relative change): large particles with low-to-moderate sinking velocity (ESD>0.8mm, SV<100m/d), medium-sized particles with moderate-to-high sinking velocity (0.3mm<ESD<0.8mm, 20m/d<SV<300m/d), and small particles with low sinking velocity (ESD<0.3mm, SV<20m/d). This classification excludes: large particles with high velocity (SV>100m/d) because inertial flow effects on nutrient transport may be substantial and, thus, the predicted degree of reshaping overestimated; medium-sized particles with low velocity (SV<20m/d) because the number of particles in this class is low; and small particles with moderate-to-high velocity (SV>20m/d) because the predicted degree of reshaping is generally low due to the small Damköhler number for such particles. Very fast sinking particles (SV>300m/d) are also excluded due to negligible plume reshaping.

Fig 5 and Fig A in S2 Appendix present the impact of the bacterial distribution and activity on plume reshaping for marine aggregates. Statistics and data for the degree of quenching per each particle class are listed in S1 Table (raw data in S1 Dataset). The general trend is that plume quenching increases with increasing particle size (higher Da) and decreasing sinking velocity (lower Pe). Large particles are often associated with lower-than-expected sinking velocities because they are characterized by low excess density, high porosity, high exopolymer (EPS) content, and low mineral content [10]. In accordance with our scaling analysis [9], the quenching of eutrophic plumes is particularly important for large particles. Uniformly distributed bacteria with normal uptake kinetics may quench the plumes by 10% to 50% around large particles, but have minimal impact (<2%) on the plumes of medium and small particles.

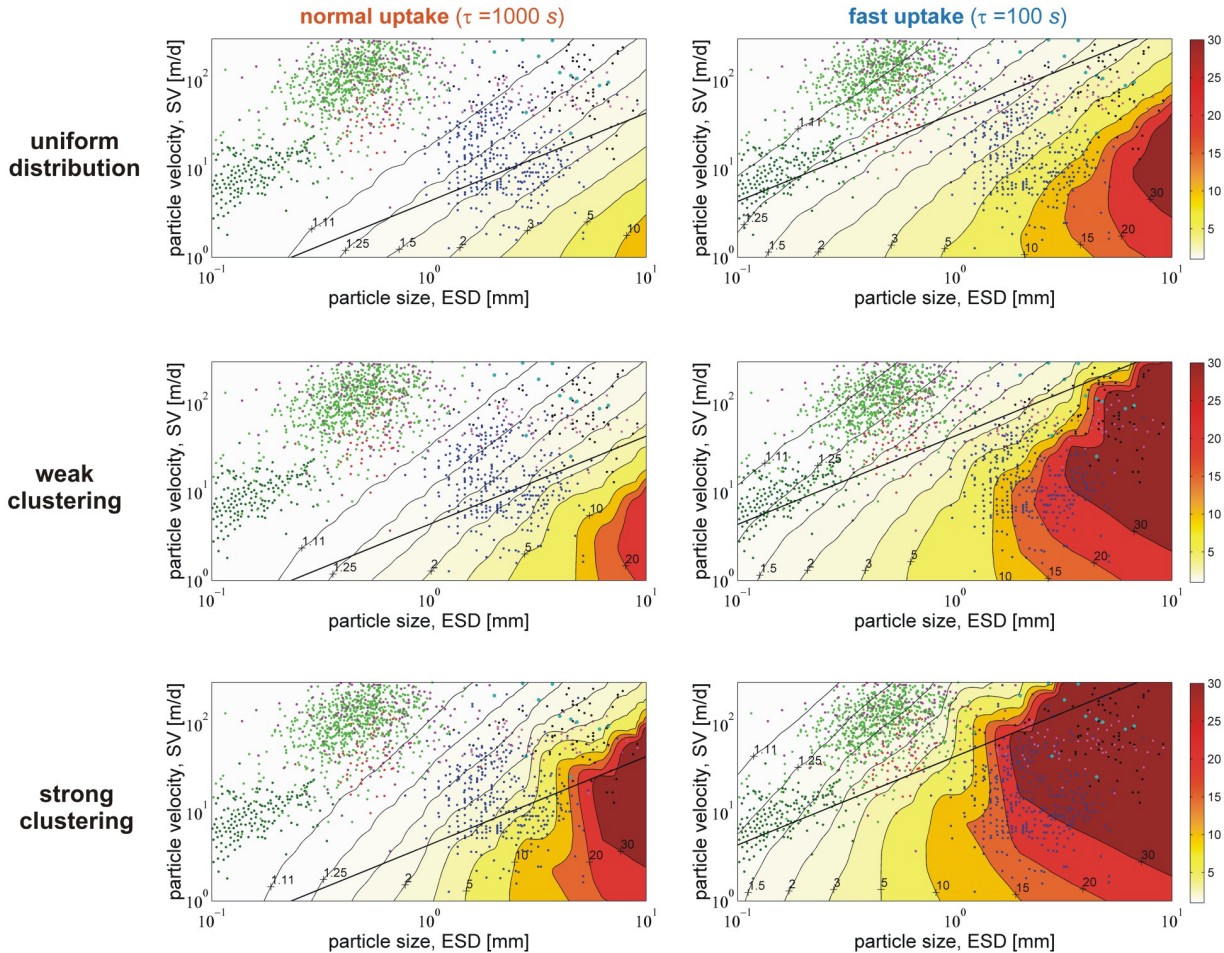

**Fig 5. Plume quenching for marine aggregates.** Predicted impact of the bacterial uptake strength and degree of clustering on the length of the trailing plume behind marine particles. The color represents the *length quenching factor*, $E_L = L^0_{plm}/L_{plm}$, which is defined as the ratio of the undisturbed plume length, $L^0_{plm}$, at zero-uptake (Da = 0) over the quenched plume length, $L_{plm}$. A quenching factor of $E_L = 2$ means that the undisturbed plume is two-times longer than the quenched, and the relative change in the plume length, $\Delta L \approx 1 - 1/E_L$, is 50%. The points represent experimental data for the sinking velocity (SV) and the equivalent sphere diameter (ESD) of individual aggregates. The straight black line corresponds to the timescale condition of Pe/Da = 100 [9]. Computations were performed for small organic solutes, like amino acids and oligo-saccharides, with a diffusivity of $\tilde{D}_{Av} = 10^{-5} cm^2/s$. The contours correspond to selected values of the quenching factor (%relative change): 1.11 (10%), 1.25 (20%), 1.5 (34%), 2 (50%), 3 (67%), 5 (80%), 10 (90%), 15 (93%), 20 (95%), and 30 (97%). Source of experimental data: [53] dark green, [54] green, [55] purple, [56] red, [57] blue, [58] magenta, [59] dark cyan, [60] black.

Bacterial clustering and elevated uptake kinetics may strikingly amplify the degree of reshaping (>90%) for plumes around large particles and, as aptly hypothesized by *Jackson* [16], make the process also significant for medium-sized and small particles (Fig 6, S1 Table). For instance, uniformly distributed bacteria with fast uptake kinetics may, on average, quench the plumes by 71% for large particles, 12% for medium and 15% for small ones. Chemotactic clustering results in a several-fold increase of the quenching effect. For example, for medium-sized particles, the average relative change in plume metrics raises from 1.4% (no clustering) to 4.2% (strong clustering) with normal uptake kinetics, while the relevant rise is from 12% (no clustering) to 33% (strong clustering) with fast uptake kinetics. Moreover, for small aggregates and phytoplankton, plume reshaping is more pronounced when considering DOM plumes of larger solutes, like proteins and polysaccharides (Fig B in S2 Appendix). The simple scaling

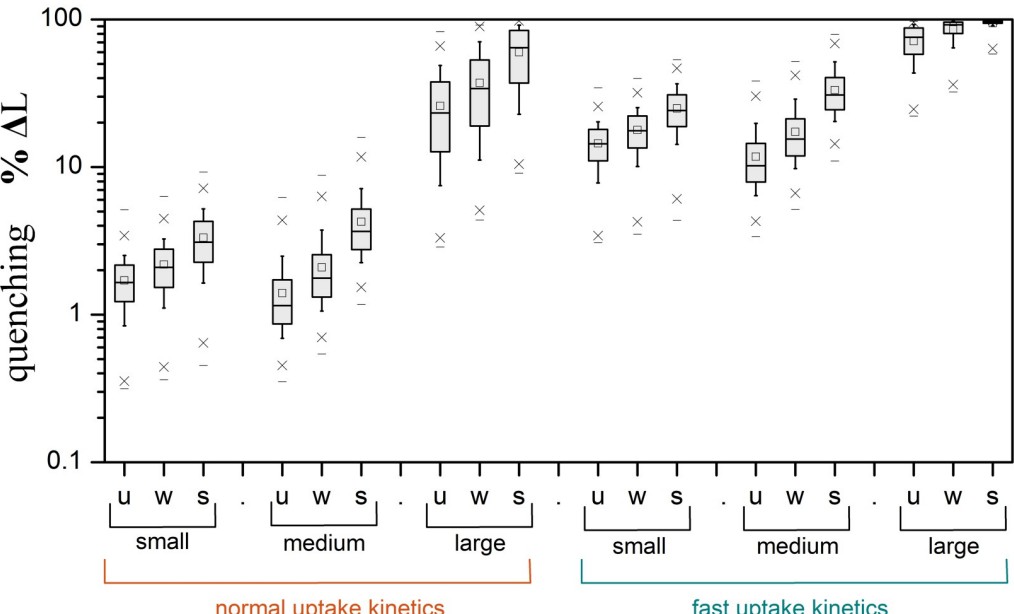

**Fig 6. Statistics of plume quenching for marine aggregates.** Box chart for the predicted impact of the bacterial uptake strength and clustering on the relative change of the plume length (%ΔL), for the three representative particle classes. Each shaded box defines the interval between the 25th and 75th percentiles, and the middle line is the median of the data. The whiskers of the box define the 10th and 90th percentiles, the square (□) is the average, the crosses (×) define the 1st and 99th percentiles, and the dashes (–) define the min/max. To create this diagram, we calculated the quenching factors for the datasets of {particle size, sinking velocity} from the experimental studies listed in the caption of Fig 5. The quenching data were sorted into bins for small, medium, and large particles in accordance to the classification of the main text. Calculations were carried out for small DOM ($\tilde{D}_{Av} = 10^{-5} cm^2/s$), normal and fast uptake ($\tilde{\tau}_U = 1000s$ or $100s$), and three levels of clustering (u = no-clustering, w = weak, s = strong; Table 1).

condition of Pe/Da<100 may be applied in an initial assessment of whether plume reshaping is relevant to a specific particle type, but its validity falls off as the degree of bacterial clustering increases. An intriguing consequence (and possible indicator) of extensive plume quenching is the detection of DOM at high concentrations within the particle and its immediate surroundings, but only at exiguous levels in ambient water [21].

To further demonstrate the potential of free-living bacteria to reshape eutrophic DOM plumes, we used as a realistic basis of our *in silico* analysis the recent work of *Alcolombri et al.* [7], who investigated the enzymatic dissolution and degradation of alginate microparticles by surface-attached bacteria of the species *Vibrio cyclitrophicus*. Alginate is a polysaccharide secreted by brown algae and serves as model marine snow. Marine bacteria colonize the particle surface and degrade the biopolymer network into oligo-alginate molecules. *Alcolombri et al.* studied particles with a radius of $\tilde{R}_P = 0.4\ mm$ and sinking velocity in the range of $\tilde{v}_\infty = 1.1 - 36.3\ m/d$. Considering a diffusion coefficient of $\tilde{D}_{Av} = 10^{-5}\ cm^2/s$ for oligo-alginate, the dissolution pattern around the microparticles is characterized by a Péclet number in the range of $Pe = 5 - 168$. The corresponding Damköhler number is $Da = 0.16$ for normal uptake and $Da = 1.6$ for fast uptake.

As shown in Fig 7, depending on the bacterial activity and distribution, nutrient uptake may cause a several-fold reduction of the plume extent. Uniformly distributed bacteria may result in measurable (>1.2-fold) quenching of the plume extent for sinking velocity $\tilde{v}_\infty < 5m/d$ ($Pe < 20$). The highest impact is caused by strong clustering of upregulated bacteria,

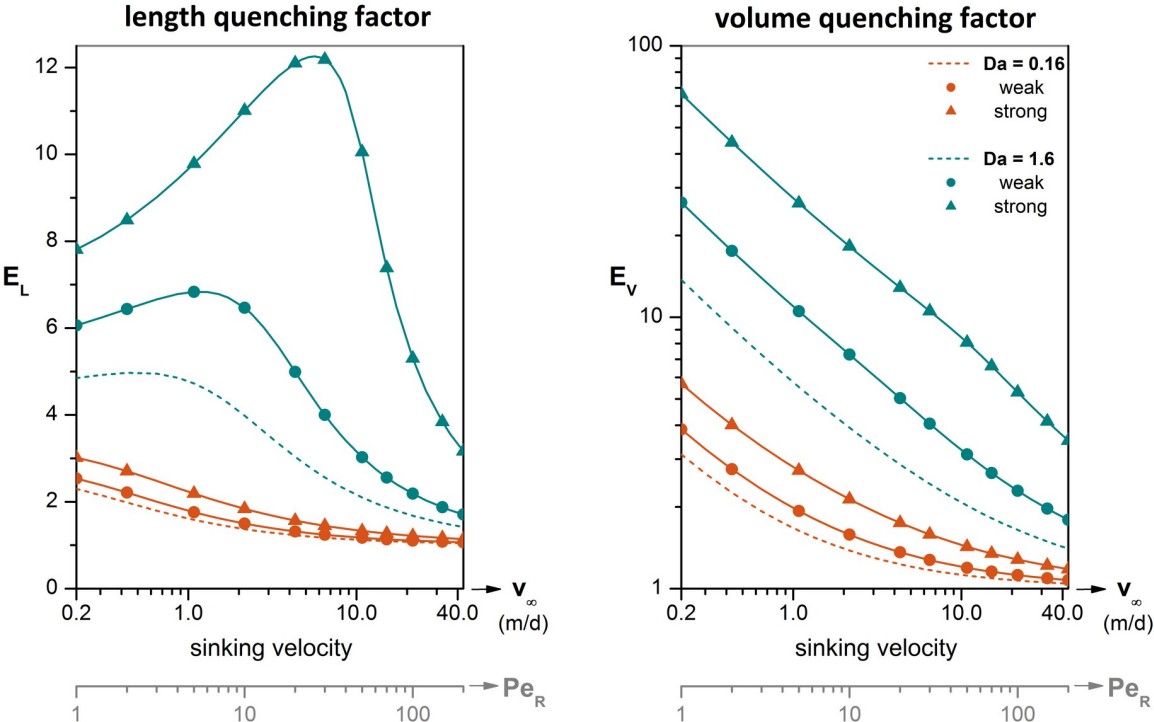

**Fig 7. Bacterial microzones amplify plume quenching.** Predicted impact of the bacterial uptake strength and degree of clustering on plume quenching factors. Similarly to the *length quenching factor*, $E_L = L_{plm}^0 / L_{plm}$, the *volume quenching factor*, $E_V = V_{plm}^0 / V_{plm}$, is the ratio of the undisturbed plume volume, $V_{plm}^0$, at zero-uptake (Da = 0) over the quenched plume volume, $V_{plm}$, at any given conditions. Undisturbed values are given in Fig C of S2 Appendix. The Damköhler is Da = 0.16 for normal uptake and Da = 1.6 for fast uptake. RDF parameters for weak and strong clustering are listed in Table 1. The Péclet number, $Pe = \tilde{R}_P \tilde{v}_\infty / \tilde{D}_{Av}$, corresponds to a particle radius of $\tilde{R}_P = 0.4mm$ and a solute diffusivity of $\tilde{D}_{Av} = 10^{-5} cm^2/s$ [7].

with a 2.6- to 12-fold quenching of the plume length and a 3.1- to 61-fold quenching of the plume volume for microparticles with sinking velocity $\tilde{v}_\infty = 0.2 - 44 m/d$ and Péclet $1 < Pe < 200$. Volumetric representations of 3D plume quenching for Pe = 5, 50 and 168 are shown in Figs D and E of S2 Appendix. The next key question is which bacterial species and under what conditions may achieve the afore-predicted levels of plume reshaping.

## Implications of plume quenching on the oceanic microbiome

Marine waters host diverse bacterial communities with trophic lifestyle ranging over the spectrum from oligotrophy to copiotrophy [61]. Typical oligotrophs, like *Pelagibacter* and *Sphingopyxis* species, are small (<0.1 μm³) non-motile cells, well adapted for slow growth in nutrient-poor waters [61–63]. Their uptake systems have high affinity and broad substrate specificity, but saturate at elevated nutrient concentrations [64,65]. Although non-motile oligotrophs are thought to drift along seawater and be homogeneously distributed around POM [21], weak clustering may occur when the hydrodynamic interactions with the particle surface are strong [37].

At the other end, copiotrophic bacteria, like *Marinobacter* and *Vibrio* species, are large cells (>1 μm³) capable of motility, environmental sensing, and thriving growth in nutrient-rich waters [61,66]. They possess multiple systems for nutrient uptake with variable affinity and substrate specificity [64,65]. Copiotrophs adapt to the changing nutrient availability in marine

waters through a feast-and-famine strategy [64,66,67]. Under prolonged scarcity of nutrients (famine), copiotrophs become idle and enter into a non-proliferating state of reduced cell size and functions (e.g., spend more than 80% of their time without swimming [68]). By contrast, during an extensive POM release (feast), copiotrophs are enabled with functions for tracking and exploiting eutrophic patches in the heterogeneous microscale seascape (e.g., rapidly boost their uptake affinity when detecting nutrient surges [52]). The energetic cost associated with the chemosensory and swimming functions of copiotrophs is compensated by the benefit of harvesting eutrophic microhabitats [28,69].

Genomic analyses have revealed that *Pelagibacters* of the SAR11 clade are prevalent in the epipelagic microbiome of the oceans [70], while the fraction of motile chemosensing bacteria is usually low (<10%) [68]. However, extensive POM releases, like algal blooms and oil spills, increase the overall bacterial abundance (>$10^7$ cells/mL) and induce the ephemeral dominance of selected copiotrophic lineages with genes for chemotaxis and energy-based uptake systems [30–32]. Microscale and trade-off analyses suggest that such transformations of the marine microbiome are sustained by the ability of copiotrophs to sense their microenvironment and outcompete oligotrophs in harvesting nutrients under the presence of particles, plumes and associated chemical gradients [21].

In this context, our results for uniformly distributed bacteria with normal uptake are relevant to *Pelagibacters* and other oligotrophs, whereas *Vibrios* and other chemotactic copiotrophs could achieve strong clustering, fast uptake and pronounced plume reshaping. Interim effects of weak clustering may be achieved by mesotrophic bacteria with reduced chemotactic attributes (e.g., the motile species *Deleya marina* is chemotactically attracted to casein, but not to valine [19]). Accordingly, copiotrophs may achieve at least 4x higher nutrient exposure than oligotrophs and 2x higher than mesotrophs (Fig 3). When the saturation effect on oligotrophic uptake is taken into account, the advantage in nutrient uptake by copiotrophs is multiplied by another 2–4 factor [9], and becomes significantly higher than previously thought. The potential impact of bacteria with different trophic lifestyles on the trailing 3D plume behind a slow-sinking particle is illustrated in Fig 8. Plume quenching may also trigger a competition of the type "first come, first served while supplies last", as successful plume trackers reduce the extent of the plume and the probability of other, less competitive, chemosensing bacteria to detect the nutrient source. To fully unravel the impacts of plume reshaping on bacterial succession dynamics during particulate blooms, our analysis could be coupled to population-based models [49].

## Outlook

In this work, we reconstructed the bacterial distribution using an exponential RDF and parameters extracted with nonlinear regression analysis of published datasets from microfluidic and computational experiments. Beyond simplicity, the developed hybrid framework has the advantage of directly coupling experimental bacterial distributions to the nutrient transport model, thus bypassing any uncertainty inherent to available bacterial transport models. Future theoretical studies could employ individual-based [26] and Keller-Segel models [28] to relate the RDF parameters to the spatial organization of bacteria around nutrient-releasing particles, by considering in detail the microscale flow and chemical fields, the bacterial chemotactic behavior, the hydrodynamic and biochemical interactions among bacteria and between the bacteria and the particle surface (cell-cell and cell-particle interactions). To this end, large datasets and intriguing bacterial distributions could also merit from advances in pattern recognition with machine learning algorithms. Particle-based models offer fundamental insight into microbial-scale processes and, ultimately, could improve the parameterization of the ocean-

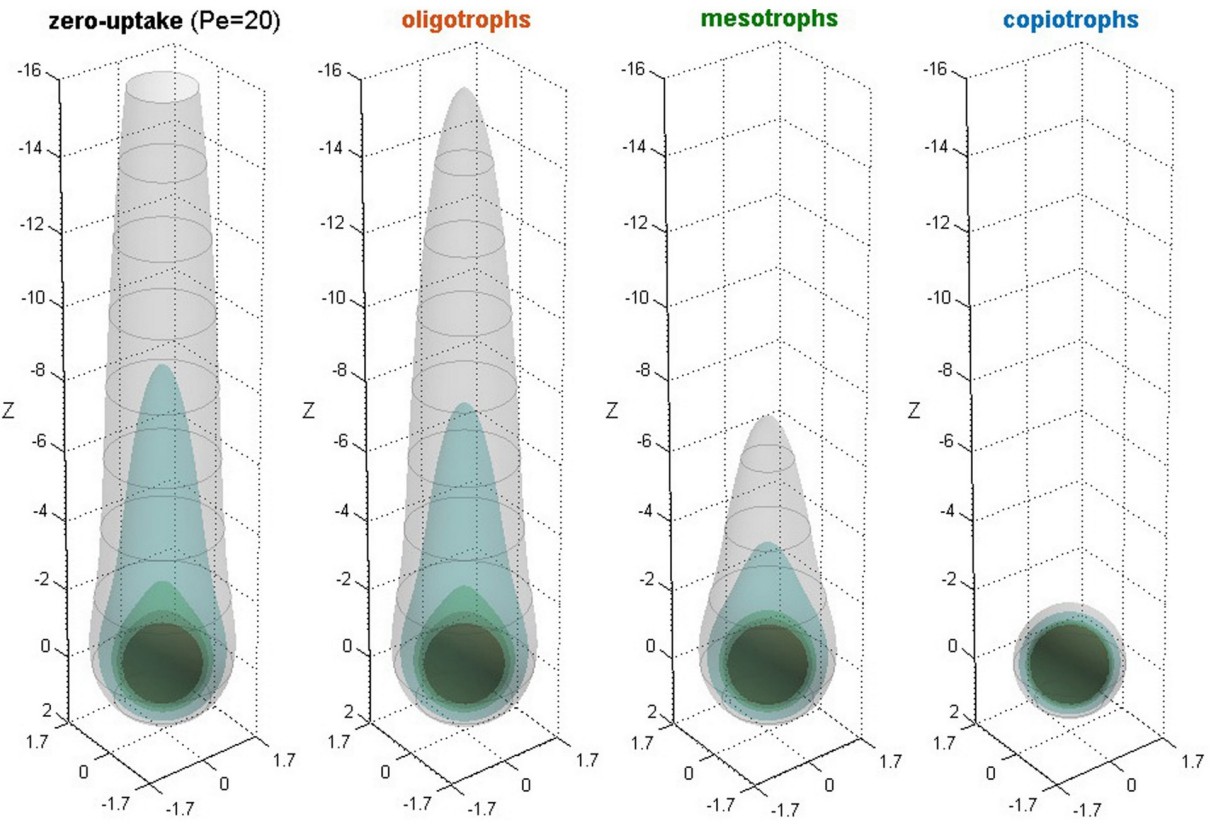

**Fig 8. Three-dimensional plume quenching.** Volumetric representation of the undisturbed nutrient plume in the wake of a slow-sinking particle (Pe = 20, Da = 0) and, also, as reshaped by oligotrophs with uniform distribution and normal uptake (Da = 0.16), mesotrophs with weak clustering and upregulated uptake (Da = 0.8), and copiotrophs with strong clustering and fast uptake (Da = 1.6). Clustering parameters are given in Table 1. Nested isoconcentration surfaces are shown at selected values of nutrient concentration ($C$ = 0.1, 0.2, 0.5 and 0.7). The Péclet number corresponds to an alginate particle of radius $\tilde{R}_P = 0.4mm$, sinking velocity $\tilde{v}_\infty = 4.4m/d$, and oligo-alginate diffusivity $\tilde{D}_{Av} = 10^{-5}cm^2/s$ [7].

level biochemical components in Earth system models, hence advancing the accuracy of predictions for POM transport, nutrient transformations and bacterial succession dynamics, especially during algal blooms, oil spills and episodic runoffs.

## Materials and methods

### Flow field around the particle

We consider a biofilm-coated organic particle as a rigid Stokes sphere of radius $\tilde{R}_P$ that moves with constant velocity through an unbounded fluid domain (Fig 1A). In the particle frame of reference, the radial and angular components of the dimensionless fluid velocity, $\mathbf{v}_v$, are:

$$v_{v,r}(r, \theta) = -\left(1 - \frac{3}{2r} + \frac{1}{2r^3}\right)\cos\theta \tag{4a}$$

$$v_{v,\theta}(r, \theta) = \left(1 - \frac{3}{4r} - \frac{1}{4r^3}\right)\sin\theta \tag{4b}$$

The above relations hold when the flow around the particle is laminar with low Reynolds number, $Re = \tilde{\rho}_v \tilde{v}_\infty \tilde{R}_P / \tilde{\mu}_v < 1$, where $\tilde{\rho}_v$ is the density of ambient water, $\tilde{\mu}_v$ is the dynamic viscosity of ambient water, and $\tilde{v}_\infty$ is the average velocity of ambient water relative to the particle. Any effect of the bacterial microzone to the flow field is assumed negligible. In the upper mixed layer, turbulence may disrupt nutrient plumes and bacterial clusters [15]. However, turbulent velocity fluctuations diminish for particles smaller than the Kolmogorov microscale, which is around 1-6mm in rough seas with high energy dissipation rate ($\sim 10^{-2}$ cm$^2$/s$^3$) [43].

## Boundary conditions on the nutrient field

Far from the particle, the nutrient concentration approaches a constant background value, $C = C_\infty$. At the particle surface, two alternative boundary conditions are considered [8]. For transport-limited dissolution, typically associated with partition equilibrium between the particulate and aqueous phases (e.g., oil-water), the solute concentration is prescribed over the particle surface, $C = C_S$. For reaction-limited dissolution, as for example associated with active exudation of metabolites by phytoplankton cells, the solute flux is prescribed over the particle surface, $-\partial C / \partial r = q_{As}$. The results presented in the paper were obtained for transport-limited dissolution with partition equilibrium at the particle-water interface ($C_S = 1$) and the nutrient field represents concentration above the background value ($C_\infty = 0$).

## Microzone radius and hotspot index

For the estimation of the degree of clustering, it is required to precisely define the extent of the bacterial microzone. In previous works, the radius of the microzone around algal cells was related to the undisturbed nutrient field (Pe = 0, Da = 0) and defined as the distance at which the nutrient concentration obtains a reference value [15,22]. Here, in relation to the bacterial RDF, we define the microzone radius as the distance at which the excess bacterial concentration obtains a reference value, that is $B(R_M) - \beta_0 = \beta_{ref}$, which results in the following expression:

$$R_M = 1 + d_s \left[ \ln \left( \beta_m / \beta_{ref} \right) \right]^{1/n} \tag{5}$$

In the calculations, we set the reference at 10% of the corresponding background value (i.e., $\beta_{ref} = 0.1$). The above definition is preferred for three reasons. First, the experimental determination of $R_M$ is straightforward. Second, the obtained $R_M$ values satisfy the notion that the microzone volume scales with the plume volume, which can be 10−100 times the particle volume [8,9] and hence give a microzone radius 2−4 times the particle radius. Finally, the resulting microzone contains more than 90% of the excess bacteria. The definition of $R_M$ enables the comparative analysis of the degree and benefits of clustering, without affecting other metrics of the nutrient field.

A metric for the degree of bacterial clustering around the particle is the hotspot index:

$$h_e \equiv \left\langle \tilde{B} \right\rangle_M / \tilde{B}_v^\infty = \frac{1}{\tilde{V}_M} \int_{V_M} B(\tilde{\mathbf{x}}) d\tilde{V} \tag{6}$$

where $\left\langle \tilde{B} \right\rangle_M$ is the average bacterial concentration within the microzone volume, $V_M$. Uniformly distributed bacteria correspond to $h_e = \beta_0$. For unbounded fluid domains, we have that $\lim_{V_M \to \infty} h_e = \beta_0$ and the consistency condition $\lim_{V_M \to \infty} \left\langle \tilde{B} \right\rangle_M = \tilde{B}_v^\infty$ demands that $\beta_0 = 1$. For confined fluid domains, it is reasonable to obtain $\beta_0 < 1$. Our results are suitable for dilute systems, with low volume fraction of POM ($< 10^{-3}$), which is a reasonable approximation for

seawater in most cases. For concentrated systems, a unit cell model that accounts for particle-particle interactions should be used [42].

## Extraction of RDF parameters

Table 1 presents values of the RDF parameters that were extracted with nonlinear regression analysis by fitting Eq (1) to available data from microfluidic and *in silico* experiments with stationary [19,21,29] or sheared particles [25]. Data collection from figures in the papers was carried out with WebPlotDigitizer, data processing was made with in-house FORTRAN algorithms, and regression analysis was performed with OriginLab. Details of the parameter extraction procedure are given in S1 Appendix.

## Plume metrics and quenching factors

The ratio of the plume volume over the particle volume is defined as

$$V_{plm} \equiv \tilde{V}_{plm}/\tilde{V}_P = \frac{3}{4\pi}\int_{V_v} H(\bar{c})dV \tag{7}$$

where $H(\bar{c})$ is the Heaviside function, with $H(\bar{c}) = 1$ if $\bar{c} > 0$ and nil otherwise, $\bar{c} = C - C_{det}$ and $C_{det}$ is the detection threshold, i.e., the minimum detectable nutrient concentration by bacteria. The length of the plume is defined as the distance from the particle center at which the nutrient concentration in the wake of the particle ($\theta = \pi$) is equal to the detection threshold, $C(L_{plm}, \pi) = C_{det}$. A detection threshold of $C_{det} = 0.1$ was used in the computations.

The degree of plume reshaping is quantified with quenching factors. The *length quenching factor*, $E_L = L^0_{plm}/L_{plm}$, is the ratio of the undisturbed plume length, $L^0_{plm}$, at zero-uptake (Da = 0) over the quenched plume length, $L_{plm}$, at any given conditions. The relative change of the plume length is related to the quenching factor as $\Delta L \equiv (L^0_{PLM} - L_{PLM})/(L^0_{PLM} - 1) \geq 1 - 1/E_L$, with $\Delta L \in [0,1]$. Accordingly, the *volume quenching factor*, $E_V = V^0_{plm}/V_{plm}$, is the ratio of the undisturbed plume volume, $V^0_{plm}$, at zero-uptake (Da = 0) over the quenched plume volume, $V_{plm}$, at any given conditions. The relative change of the plume volume is related to the quenching factor as $\Delta V \equiv (V^0_{PLM} - V_{PLM})/V^0_{PLM} = 1 - 1/E_V$. A quenching factor of 1.25 corresponds to a relative change of 20%, and a quenching factor of 10 corresponds to a relative change of 90%.

## Efficiencies of dissolution, degradation and plume exploitation

The total nutrient flux through a spherical surface with radius $r$ from the center of the particle is defined as:

$$\mathcal{Q}_{As}(r) \equiv \tilde{\mathcal{Q}}_{As}(\tilde{r})/\left(\tilde{q}_{ref}\tilde{R}^2_P\right) = \int_S \mathbf{q}_A \cdot \mathbf{e}_r dS \tag{8}$$

where $\tilde{q}_{ref} = \tilde{C}_{ref}\tilde{D}_{Av}/\tilde{R}_P$ is the reference flux, $\mathbf{q}_A = \text{Pe}\mathbf{v}C - \nabla C$ is the *combined* nutrient flux that accounts for both advection and diffusion, and $dS = r^2 \sin\theta d\theta d\varphi$ is the differential area on the spherical surface. The Sherwood number, $Sh_R = \tilde{\mathcal{Q}}_{As}(\tilde{R}_P)/\left(4\pi\tilde{R}_P\tilde{D}_{Av}\tilde{C}_{ref}\right)$, represents the ratio of the total nutrient flux, inclusive of advection and consumption effects, over the diffusive nutrient flux alone. The dissolution enhancement that is caused by nutrient consumption is calculated as $E_{dis} = Sh_R/Sh_{R0}$ where $Sh_{R0}$ is the Sherwood number in the absence of consumption (Da = 0). The degradation efficiency is the fraction of released nutrient that is consumed within a spherical shell of outer radius $r$, $E_{deg}(r) = 1 - \mathcal{Q}_{As}(r)/\mathcal{Q}_{As}(1)$.

### High-resolution numerical scheme for nutrient transport

The advection-diffusion-bioreaction equation given in Eq (2) is solved numerically with a finite difference scheme, which is described in detail in [9]. Briefly, the two-dimensional $(r,\theta)$ space is discretized with a body-conforming non-uniform grid. The grid density is high around the particle and also downstream so as to capture the large concentration gradient in these areas. The advection operator is discretized with a third-order upwind scheme, whereas the diffusion operator with central differences. The numerical results are grid independent and the accuracy has been confirmed by comparison with literature data and correlations [9].

## Supporting information

**S1 Dataset. Raw data for statistics in S1 Table and Fig 6.**
(ZIP)

**S1 Appendix. RDF extraction.**
(PDF)

**S2 Appendix. Supporting figures.**
(PDF)

**S1 Table. Characteristics of particle classes.**
(PDF)

## Acknowledgments

We are grateful to Roman Stocker (ETH Zürich) for insightful discussions at the beginning of this project and to the Fields Institute for overall support through the Thematic Program on Emerging Challenges in Mathematical Biology.

## Author Contributions

**Conceptualization:** George E. Kapellos, Hermann J. Eberl, Nicolas Kalogerakis, Patrick S. Doyle, Christakis A. Paraskeva.

**Data curation:** George E. Kapellos.

**Funding acquisition:** George E. Kapellos, Nicolas Kalogerakis, Patrick S. Doyle.

**Methodology:** George E. Kapellos, Hermann J. Eberl, Christakis A. Paraskeva.

**Resources:** Hermann J. Eberl, Nicolas Kalogerakis, Patrick S. Doyle, Christakis A. Paraskeva.

**Supervision:** Nicolas Kalogerakis, Patrick S. Doyle, Christakis A. Paraskeva.

**Validation:** Hermann J. Eberl.

**Writing – original draft:** George E. Kapellos.

**Writing – review & editing:** George E. Kapellos, Hermann J. Eberl, Nicolas Kalogerakis, Patrick S. Doyle, Christakis A. Paraskeva.

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
