## [Decision Letter · Decision Letter 0]

23 Dec 2023

Dear Dr Kapellos,

Thank you very much for submitting your manuscript "Bacterial clustering amplifies the reshaping of eutrophic plumes around marine particles: a hybrid data-driven model" for consideration at PLOS Computational Biology.

Your manuscript was reviewed by members of the editorial board and by an independent reviewer. In light of the review (below this email), we would like to invite the resubmission of a significantly-revised version that takes into account the reviewers' comments.

In addition, please ensure that all data and code required to reproduce the code should be make available. "Additional data related to this paper may be requested from the authors" is not adhering to the PLoS Comp Biol principles. Please make the code and data available in an open source repository such as GitHub. 

We cannot make any decision about publication until we have seen the revised manuscript and your response to the reviewers' comments. Your revised manuscript is also likely to be sent to reviewers for further evaluation.

Sincerely,

Kiran R. Patil, Ph.D.

Section Editor

PLOS Computational Biology

Reviewer's Responses to Questions

**Comments to the Authors:**

Reviewer #1: The manuscript presents an analysis of microbial quenching of nutrient plumes that form from particulate matter in marine environments. It builds on the model framework and analysis described in a previously published article (Microorganisms 2022) by the authors. In the original paper the bacterial distribution was considered uniform. In this new work the microbial population is modeled using a radial distribution function (RDF), which lumps together flow and chemical fields, chemotaxis, hydrodynamic interactions and particle-surface interactions. The RDF is based on data (experimentally and computationally derived) that were mined from existing literature to fit model parameters. An advection-dispersion-reaction model is used to describe nutrient concentration in the plume. The effect of two factors on plume quenching, clustering due to chemotaxis and fast nutrient uptake, were quantified in phase diagrams.

Qualitatively, it makes sense that greater clustering and slower uptake would impact the plume to the greatest extent. However, the quantitative analysis is critical for truly gauging its relevance to microbial impact on the nutrient plumes from settling particles. The analysis suggests that only under conditions of strong clustering and fast nutrient uptake do the microorganisms impact the extent of the plume. The authors make the case that copiotrophs (e.g. Vibrio) operate under such conditions. The authors related different parametric regimes (particle size, sinking velocity, uptake rate and clustering) to specific marine microorganisms and the extent of plume quenching. I found this aspect to be a significant strength of the manuscript.

Technical comments

1. In Fig 1B the RDF model for strong clustering overestimates the bacterial concentration for all the data and the weak clustering model overestimates the Barbara data and a good portion of the Desai (r>1.5) and Smirga (r>2.2) data. To what extent does this then skew the stated importance of microbial reshaping of the plume?

2. Table 1 – Parameters for the algal cell (Bowen) seem to be outliers compared to the other 3 data sets listed. Are they consistent with known mechanistic differences for the plume formation? For example, does it make a difference that the algae are actively swimming rather than sinking.

3. I would expect the functional form and values of the fitting parameters for the RDF model to provide insight to the underlying processes that are dominating the bacterial distributions in the microzone. Please elaborate on any insight gained from determining the exponential functional dependence and magnitude of the parameters in the RDF model. For example, does a larger value of n imply a stronger chemotactic response? If so, is that consistent with the strength of the chemotactic response for the algal cell in the Bowen work compared to the other sources?

4. In Fig 3, provide more guidance for the reader to interpret the information presented in this phase diagram. It appears that only large particles with low-to-moderate sinking velocity and fast uptake kinetics are susceptible to plume quenching by marine microorganisms.

Editorial comments

1. L24 As written, this statement does not convey that the high potential for reshaping is limited to conditions of higher than normal uptake, which are typically associated with copiotrophs under a feasting scenario. Please rephrase to indicate that the high potential for reshaping applies to only certain conditions.

2. L97-99 Cite the sources of the data sets that are mentioned.

3. L157 & 161 Include that C is also a function of x.

4. L180 Is the plume lifetime range from several seconds to tens of minutes rather than tenths of minutes? As stated, plume lifetimes from several seconds (e.g. 3 s) to tenths of minutes (e.g. 0.1 x 60s/min = 6 s) are nearly comparable values.

5. L185 Please clarify how chemotaxis is included in the terms for Eq (1). Does chemotaxis (clustering) increase BM and decrease dS? Then the hotspot metric is derived from these parameters using B(x). Is that correct? Or, do you start with an expected value for the hotspot metric and infer the parameter values for Eq (1)?

6. L188-191 The given ranges for the average bacterial abundance and affinity yield uptake timescales that range from 10^2 s to 10^10 s. Perhaps, the authors meant for the text to read tens of femtoliters rather than tenths of femtoliters. Even so, these ranges do not yield times of less than 100 s, so the statement that the uptake timescale may range from tenths to hundreds of seconds is incorrect. I noticed that this error was carried over from a previous publication (Microorganisms 2022).

7. L192 Cite the source for the recently shown result.

8. L215 Clarify what is meant by essential in the context, “particle classes for which plume reshaping is essential”.

9. L216 In categorizing the particle classes, if low-to-moderate sinking velocity is classified as SV<100m/d, did the authors intend for moderate sinking velocity to be SV>100m/d? Otherwise, a large portion of the medium particles lie outside the criterion.

10. L233-236 For which uptake strength does this description apply? It does not seem consistent with the panels in Fig 3. For example, for uniform distribution and fast uptake the medium sized particles mostly lie outside the EL=2 (50%) contour. Please clarify the description of the bounds for which quenching is important.

11. L243-245 Cite the source for this statement.

**Have the authors made all data and (if applicable) computational code underlying the findings in their manuscript fully available?**

Reviewer #1: Yes

PLOS authors have the option to publish the peer review history of their article (what does this mean?). If published, this will include your full peer review and any attached files.

Reviewer #1: No
---

## [Decision Letter · Decision Letter 1]

19 Nov 2024

Dear Dr Kapellos,

We are pleased to inform you that your manuscript 'Bacterial clustering amplifies the reshaping of eutrophic plumes around marine particles: a hybrid data-driven model' has been provisionally accepted for publication in PLOS Computational Biology.

Best regards,

Marc R Birtwistle, PhD

Section Editor

PLOS Computational Biology

Kiran Patil

Academic Editor

PLOS Computational Biology

Feilim Mac Gabhann

Editor-in-Chief

PLOS Computational Biology

Jason Papin

Editor-in-Chief

PLOS Computational Biology

Reviewer's Responses to Questions

**Comments to the Authors:**

Reviewer #1: I appreciate the care and thoroughness with which the authors addressed my comments. Well done!

Reviewer #2: This study examines how microbes consume nutrient plumes released from particles in the ocean. This research considers microbes clustering around particles, influenced by factors like flow, chemistry, and surface interactions. The model uses data from experiments and simulations to capture these dynamics using an RDF and explores how nutrient levels in plumes change due to microbial activity, focusing on the effects of clustering and rapid nutrient consumption.

I do not have any major objections to the methodology (I leave that to other reviewers with more expertise on fluid dynamics). I will try instead to describe what I interpret to be the major contribution of this work in the context of marine microbiology and carbon cycling. From that standpoint, it seems to me the major contribution of this paper is providing a mathematical framework to define the interplay between microbes and nutrient patches in an efficient manner, which can further facilitate efforts to "upscale" from detailed processes at the single-cell level to processes at the particle (and eventually water column) scale. I can imagine for example how the use of RDFs as detailed in this study can be applied to model carbon fluxes at large scales, as for example done in Nguyen, Zakem et al, 2022, Nature Communications, but using a more efficient representation of the micro- and meso-scale processes. In the future, it would be useful to show how the RDF parameters could change throughout the water column and what the impact of these changes would be on carbon transport and recycling by microbes. With that in mind, I believe this is a valuable contribution to the field.

**Have the authors made all data and (if applicable) computational code underlying the findings in their manuscript fully available?**

Reviewer #1: Yes

Reviewer #2: None

PLOS authors have the option to publish the peer review history of their article (what does this mean?). If published, this will include your full peer review and any attached files.

Reviewer #1: No

Reviewer #2: No

---

## [Editor Report · Acceptance letter]

28 Nov 2024

PCOMPBIOL-D-23-01593R1 

Bacterial clustering amplifies the reshaping of eutrophic plumes around marine particles: a hybrid data-driven model

Dear Dr Kapellos,

I am pleased to inform you that your manuscript has been formally accepted for publication in PLOS Computational Biology. Your manuscript is now with our production department and you will be notified of the publication date in due course.

With kind regards,

Zsofia Freund
